# Small but threatening: Punctate Outer Retinal Toxoplasmosis (PORT), a case series report

Rebeca Paquentín-Jiménez, Ronald Rivera-Sempértegui, Luz Elena Concha-Del-Río  *

Inflammatory Eye Disease Clinic, Asociación Para Evitar la Ceguera en México I.A.P, Mexico City, Mexico

* luzelena.concha@comunidad.unam.mx, luzelena.concha@apec.com.mx

## Abstract

### Aim

To emphasize the importance of promptly assessment of punctate outer retinal toxoplasmosis (PORT), and to present a case series in a third-level reference center in Mexico City.

### Methods

Retrospective, observational case series of eight eyes of eight patients who presented PORT between January 2020 to February 2023. Diagnosis was made by uveitis specialists based on clinical findings, serology, and multimodal imaging. Descriptive statistics were used to exhibit absolute and relative frequencies.

### Results

Patient ages ranged from 19 years to 67 years old. Each patient presented creamy yellowish-white, punctate outer retinal lesions. All patients had SD-OCT showed outer retinal alterations and in autofluorescence presented hyperautofluorescent lesions. 85% of the patients improved their BCVA after treatment, but three (42.8%) were legally blind.

### Conclusions

PORT is an atypical presentation of ocular toxoplasmosis. Lesions usually lead to retinochoroidal scarring which can significantly diminish a patient's BCVA irreversibly. Recognition of PORT lesions is important since timely and efficacious therapy leads to better visual prognosis and diminish the risk of legal blindness.

**Data availability statement:** All data are in the manuscript and https://doi.org/10.5281/zenodo.15116904

**Funding:** The author(s) received no specific funding for this work.

**Competing interests:** I have read the journal's policy and the authors of this manuscript have the following competing interests:RP-J and RR-S have declared that no competing interests exist. LEC-d-R has received speaker honorarium and financial support for educational programs from Abbvie and Janssen.

## Author summary

Uveitis comprises a broad group of intraocular inflammatory diseases that can be infectious or non-infectious. Together, they can cause up to 20% of legal blindness in first-world countries and even more in developing countries. Ocular toxoplasmosis is the most common cause of infectious posterior uveitis. Atypical presentations of ocular toxoplasmosis, such as PORT, may often be misdiagnosed, which can cause severe ocular complications that negatively impact the patient's quality of vision and life. The aim of this paper is to emphasize that ocular toxoplasmosis may present in typical or atypical variants. Even though atypical presentations are less frequent, they can be equally or even more dangerous than the classic variant if not diagnosed and treated promptly.

## Introduction

Ocular toxoplasmosis (OT) is the most common cause of infectious posterior uveitis. [1,2] It can be either a congenital or an acquired disease. [1] Clinical presentations can be classical or atypical. The classical presentation consists of a full-thickness necrotizing retinochoroiditis focus with moderate to severe vitreous inflammation. [3–7] Atypical presentations of OT can occur either as an initial manifestation or in combination with the classical form. These atypical manifestations include scleritis, multifocal retinochoroiditis, exudative or rhegmatogenous retinal detachment, isolated retinal vasculitis, pigmentary retinopathy, neuroretinitis, papillitis, retinal vein occlusion, punctate inner retinal toxoplasmosis (PIRT) and punctate outer retinal toxoplasmosis (PORT). [3,8]

Gass identified an unusual toxoplasmosis affecting outer retinal layers in 1968. [9] A year later, Friedmann and Knox reported similar cases with these findings and with mild-to-none vitreous inflammation. [10] In 1985, Doft and Gass introduced PORT as a term. [11]

PORT is characterized by multifocal grey-white lesions at the outer retinal layers and retinal pigment epithelium associated with minimal vitreous inflammation. [1,3,4,9–11] Atypical presentations are more frequent in patients with systemic immunosuppression. [3] Most lesions have been reported in the macula, but this may be because macular lesions are visually the most symptomatic. [1,4] Recurrences when they appear, usually involve the adjacent retina. [2,5–7] Complications encountered include choroidal neovascularization, serous retinal detachment, [4] and macular cysts[7] that may convert to what is known as huge outer retinal cystoid space (HORC). [12] Treatment with pyrimethamine, sulfadiazine, and corticosteroids often results in full visual recovery, yet retinochoroidal scarring is often seen with optical coherence tomography (OCT). [3]

The aim of the study is to describe clinical findings, serology, multimodal imaging, and visual prognosis in patients with PORT to facilitate further recognition of this pathology and thereby decrease the frequency of irreversible complications leading to legal blindness.

## Methods

### Ethics statement

All information for this study was obtained from medical records, ensuring no risk to our patients. The study followed the guidelines of the Declaration of Helsinki and was approved by the institutional review board of Asociacion Para Evitar la Ceguera ethics and research committee (UV-18–01). Written consent was given by the patients for their information, and it is stored in the medical record of the hospital database and used for research. The data were analyzed anonymously.

We retrospectively analyzed cases with OT diagnosis from January 2020 to February 2023 at Asociación Para Evitar la Ceguera en México I.A.P, a third-level reference center in Mexico City. Data from 498 patients with toxoplasmosis infection were obtained from the electronic medical records. Diagnosis of toxoplasmosis uveitis was made based on clinical features and positive serology (anti-Toxoplasma Immunoglobulin G (IgG). Additional laboratory exams such as tuberculin skin test (TST), Venereal Disease Research Laboratory (VDRL) test and Fluorescent Treponemal Antibody Absorption (FTA-Abs) were also obtained to rule out other causes of retinochoroiditis. Patients were evaluated using available images including OCT, fundus photography, and fundus autofluorescence (FAF). All imaging studies were performed by RRS.

PORT cases were selected from the patients with other OT presentations. Cases with insufficient documentation were excluded. PORT diagnosis was made by a uveitis specialist based on clinical findings, positive serology, and adequate treatment response (800 mg sulfamethoxazole and 160 mg trimethoprim every 12 hours, clindamycin 300 mg every 6 hours and tapering steroids initiating with 0.5 mg/kg/day of prednisone). Patient data was captured in Microsoft Excel spreadsheets and processed with SPSS version 23. Descriptive statistics were used to exhibit absolute and relative frequencies.

## Results

From 498 medical records with OT, eight patients with PORT were recovered. (**Table 1**) Seven (87.5%) of the eight patients were female. The median age of presentation was 30.5 years old (IQR 23 – 57.5). All cases were unilateral. The median time patients presented symptoms before consultation was 30 days (IQR 15 – 150).

The median best corrected visual acuity (BCVA) was -0.85 logMAR (~20/125) (IQR -0.55 to -1.75) at the initial presentation and -0.30 (~20/40) (IQR -0.20 to -1.3) at the final visit. Thus, six out of seven eyes (85%) improved their vision after treatment. The final BCVA of patient #7 was not retrieved since the patient did not return, therefore this data was not considered. All lesions were superficial, minimally elevated, with a yellow-white creamy appearance and minimal associated vitritis. An overview of the eight patients is presented below.

### Case 1

Female in her seventh decade of life presented with decreased vision and metamorphopsias in OS (oculus sinister) for 5 days. Her BCVA in the OD (oculus dexter) was 20/20 and 20/800 in the OS. Intraocular pressure (IOP) was 11 mmHg and 14 mmHg in OD and OS respectively. OD examination was normal. OS had no anterior chamber inflammation; however, she presented 2 + vitritis and a yellowish-white foveal lesion. Spectral-domain optical coherence tomography (SD-OCT) showed a 494 µm macular hole, epiretinal membrane (ERM), and outer layer retinal hyperreflectivity, corresponding to a PORT lesion. The presumptive diagnosis was made, however, the patient returned one month later, with a positive enzyme-linked immunosorbent assay (ELISA) for Toxoplasma IgG (>650 IU/ml, reference value <30) and IgM (18.9 IU/ml, reference value <1 IU/ml). Treatment with sulfamethoxazole/trimethoprim (SMZ/TMP), clindamycin, and tapering oral prednisone was initiated and completed after 2 months. Fourteen months later, BCVA was 20/200 in the OS. IOP was 13 mmHg in OU (oculus uterque). Clinically, a retinochoroidal atrophic scar was demonstrated, and no signs of disease activity were observed. SD-OCT exhibited a healed macular hole, and a central area of full-thickness retinal layer disorganization, and ERM **Fig 1** demonstrates the SD-OCT follow-up of this patient.

**Table 1. Patient clinical characteristics among follow-ups.**

| Patient | Age/Gender | Laterality | Location of PORT | Initial BCVA | Final BCVA | Initial IOP | Final IOP | First episode or reactivation/Disease activity | Recurrence | Treatment |
|---|---|---|---|---|---|---|---|---|---|---|
| 1 | 66/F | OS | Foveal | 20/800 | 20/200 | 11 | 13 | First episode/Active | No | SMZ/TMP + DA + PRED |
| 2 | 31/F | OS | Foveal | CF | 20/40 | 24 | 12 | First episode/Active | Yes | SMZ/TMP + DA + PRED + Topical prednisolone + Dorzolamide and timolol |
| 3 | 32/M | OD | Foveal | 20/100 | 20/400 | 14 | 14 | First episode/Active | No | SMZ/TMP + PRED |
| 4 | 67/F | OD | Foveal | CF | 20/800 | 10 | 12 | Inactive | No | Intravitreal aflibercept |
| 5 | 22/F | OD | Macular. Superior temporal | 20/60 | 20/40 | 12 | 14 | Reactivation*/Active | No | SMZ/TMP + DA + PRED |
| 6 | 26/F | OD | Perifoveal | 20/40 | 20/20 | 28 | 13 | First episode/Active | No | SMZ/TMP + DA + PRED + Dorzolamide and timolol |
| 7 | 19/F | OD | Macular. Temporal | 20/200 | - | 13 | - | Reactivation*/Active | - | SMZ/TMP + DA + PRED |
| 8 | 29/F | OS | Perifoveal | 20/100 | 20/30 | 14 | 18 | Reactivation*/Active | No | SMZ/TMP + DA + PRED + Topical prednisolone |

CF: counting fingers; DA: clindamycin; F: female; M: male; OD: oculus dexter; OS: oculus sinister; PRED: Oral Prednisone; SMZ/TMP: Trimethoprim/sulfamethoxazole.

*Full thickness choroidal scar.

## Case 2

Female in her fourth decade of life attended with a diminution of vision in her OS for 6 months. Her BCVA was 20/25 in OD and CF (counting fingers) in OS. IOP were 15 mmHg and 24 mmHg in OD and OS respectively. OD clinical examination presented no abnormalities. OS presented 1 + cells in the anterior chamber, 1 + cells in the vitreous, and a yellowish-white creamy retinal lesion at the fovea. SD-OCT demonstrated macular ERM, outer retinal lesions, and ellipsoid disruption. Laboratory testing revealed elevated toxoplasma IgG (73.6 UI/mL, reference value >8.8 IU/ml), and IgM was negative. Systemic treatment with SMZ/TMP, clindamycin, and tapering oral prednisone was established. OS was also treated locally for anterior chamber inflammation and ocular hypertension with topical prednisolone and dorzolamide/timolol. Gradual resolution of the retinochoroiditis lesion was observed along the onset of pigmentary changes at the macula. After treatment termination at 2 months, her OS BCVA was 20/100 and the IOP was 18 mmHg. Six months later, the patient returned with a deterioration of visual acuity in the OS. Clinical examination of OD remained unremarkable. The OS examination exhibited a BCVA of 20/150 and an IOP of 10 mmHg. Anterior chamber presented 2 + cells, and fundus examination, revealed a new white lesion adjacent to the earlier PORT lesion at the fovea. FAF also demonstrated a hyperautofluorescent lesion, corresponding with the clinically observed lesion. Thus, the previous treatment was restored for 2 months. A stable scar at the macula was seen. Her final BCVA after treatment in the OS was 20/40, and her IOP was 12 mmHg. Fundus photography, SD-OCT, and AF follow-up of this patient are observed in **Fig 2**.

## Case 3

Male in his fourth decade of life presented with myodesopsias and a progressive decrease in visual acuity in the OD in the past 2 months. BCVA and IOP were 20/100 and 12 mmHg in the OD, and 20/20 and 14 mmHg in the OS. OS clinical examination was normal. OD clinical examination revealed 1 + of vitreous inflammation and a foveal yellowish-white lesion. SD-OCT exhibited disorganization of outer retinal layers. FAF revealed a foveolar hyperautofluorescent lesion. The patient returned after one month with positive IgG (>450 IU/ml, reference value <10.5 IU/ml) and negative IgM. Treatment with

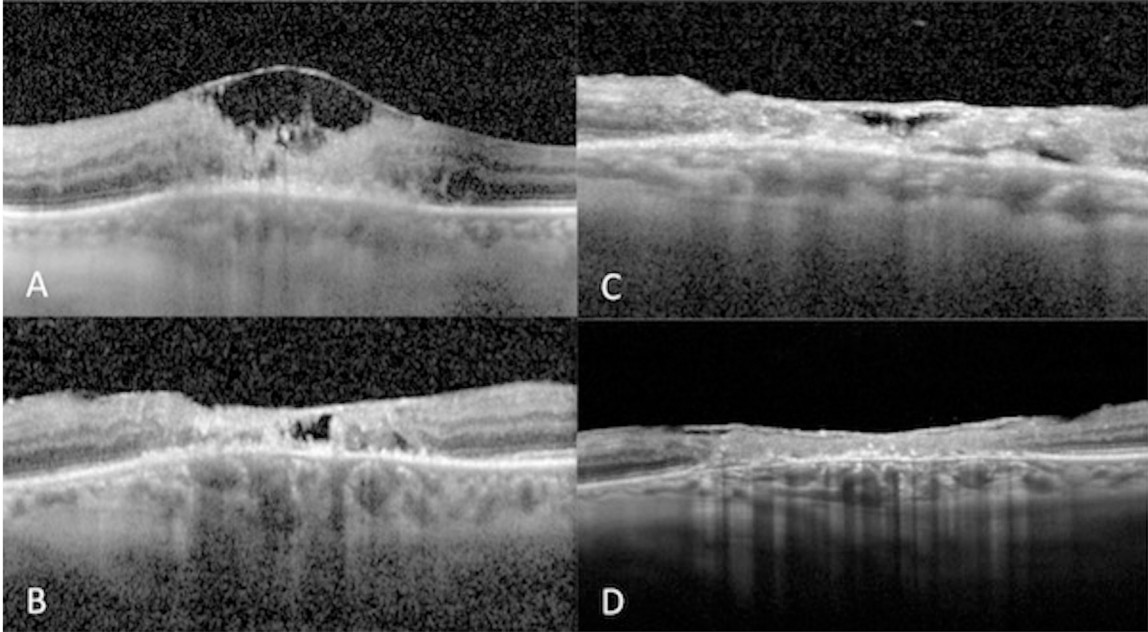

**Fig 1. Left-eye SD-OCT of patient 1 throughout disease evolution. A.** Attached vitreous at the level of the vitreoretinal interface with punctate hyper-reflective vitreous cells, a loss of foveal architecture, an epiretinal membrane, disruption of inner retinal layers and hyper-reflective material within outer retinal layers. **B.** Full-thickness involvement corresponding to retinochoroidal lesion. **C.** epiretinal membrane, central irregular hyper-reflective material within the zone of RPE atrophy. **D.** Macular atrophy with an epiretinal membrane, hyper-reflective points within outer retinal layers, and irregularity of RPE.

SMZ/TMP, clindamycin, and tapering oral prednisone was initiated. Three months later, after completing the treatment, the patient returned clinically inactive, with a final OD BCVA of 20/400, and IOP was 14 mmHg. SD-OCT demonstrated macular atrophy due to scarring and an ERM. AF showed a rounded hyperautofluorescent lesion adjacent to the fovea. (**Fig 3**) Up to December 2022, the patient has been inactive for 7 months.

## Case 4

Female in her eighth decade of life attended our hospital with a slow progressive diminution of vision in her OD 7 months ago. BCVA was CF in her OD, and 20/30 in her OS. IOP was 10 mmHg in OU. Clinical examination in OS was unremarkable. OD examination exhibited a macular greyish scar with an adjacent yellowish-white lesion and a site of subretinal haemorrhage. Toxoplasma IgG was elevated (98.2 IU/mL, reference value >8.8 IU/ml), and IgM was negative. SD-OCT exhibited intraretinal cysts, outer retinal layer disorganization, focal disruption of the ellipsoid, interdigitation zone (IZ), and a hyperreflective subfoveal image corresponding to a choroidal neovascular membrane. (**Fig 4**) Due to this, an anti-vascular endothelial growth factor (aflibercept) intravitreal injection was prescribed for her OD. Post-injection BCVA in the OD was 20/800 and IOP was 12 mmHg. SD-OCT revealed improvement of macular architecture, diminution of cyst size, chorioretinal atrophy, and subfoveal fibrosis. (**Fig 4**) The patient was scheduled for further follow-up; however, she failed to return to our hospital.

## Case 5

Female in her third decade of life presented with blurred vision in her OD for one month. BCVA was 20/60 in her OD and 20/20 in her OS. IOP was 12 mmHg in OU. The ophthalmological examination of her OS was normal. OD presented 1 + of

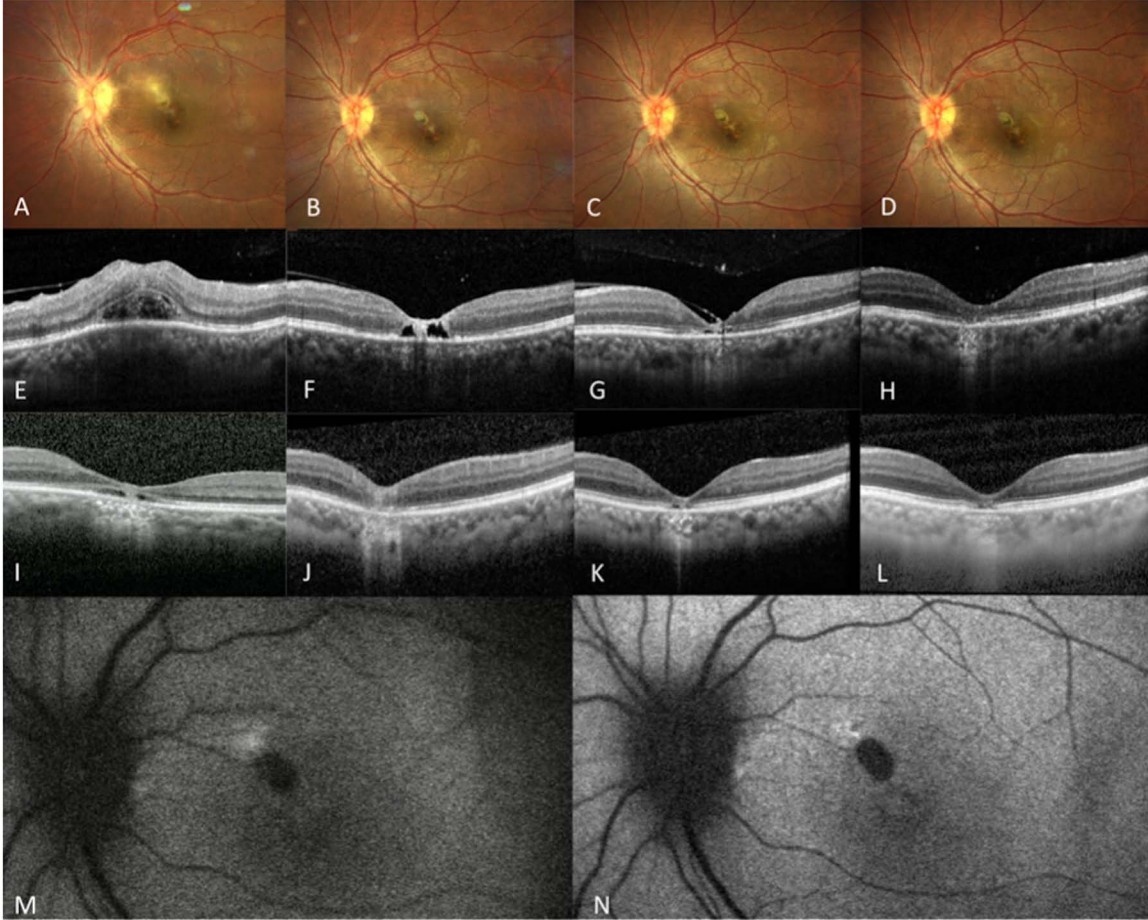

**Fig 2. A-D. Color fundus photography follow-up of patient 2. A.** Left eye color fundus photography at initial presentation showing disc pallor, an active creamy yellowish PORT lesion in the parafoveal area. **B.** Disc pallor improvement, PORT lesion starts becoming hyperpigmented. **C and D.** Disc pallor is no longer notable, and the PORT lesion is reduced in size and appears more pigmented at two months of follow-up. **E-L.** SD-OCT scan follow-up of the same patient. **E.** SD-OCT scan demonstrating vitreous cells, foveal architectural loss, subretinal fluid pocket with white material accumulations and disruption of the ellipsoid zone. **F.** Atrophy of outer retinal layers. **G.** Thinning of outer retinal layers. **H.** Disruption of the ellipsoid zone and atrophy of RPE. **I.** Foveal thinning, subretinal disruption with hyper-reflective material in outer retinal layers, and backscattering. **J.** Foveal atrophy, ellipsoid disruption, RPE atrophy, and backscattering. **K.** Important foveal atrophy, ellipsoid zone disruption and backscattering. **L.** Atrophic scarring at the fovea with central loss of the ellipsoid zone and hyperreflective backscattering. **M-N.** Fundus autofluorescence follow-up of the same patient. **M.** Hypo-autofluorescence area in the perifovea corresponding to PORT lesion with a hyper-autofluorescent superonasal edge. **N.** At the 10-month follow-up, shows a hypo-autofluorescence in the perifovea corresponding to the PORT lesion with a hyper-autofluorescent superonasal edge and a mottled hyper-autofluorescence between the disc and the PORT lesion.

vitreous cells, a superior-temporal yellowish-white lesion in the macula, and two adjacent hyperpigmented rounded retino-choroidal scars. (**Fig 5**) SD-OCT revealed multiple resolved chorioretinal lesions, a recent intraretinal lesion affecting outer layers, ERM, and tortuous choroidal vessels. Toxoplasmosis IgG was positive (6.7 IU/ml, reference value <3 IU/ml) and IgM was negative, therefore a reactivation of PORT was diagnosed. Treatment was initiated with SMZ/TMP, clindamycin, and tapering oral prednisone. Two months later, after completing the treatment, the final BCVA in OD was 20/40, however, the patient complained about having a scotoma in her central visual field. The final IOP was 14 mmHg.

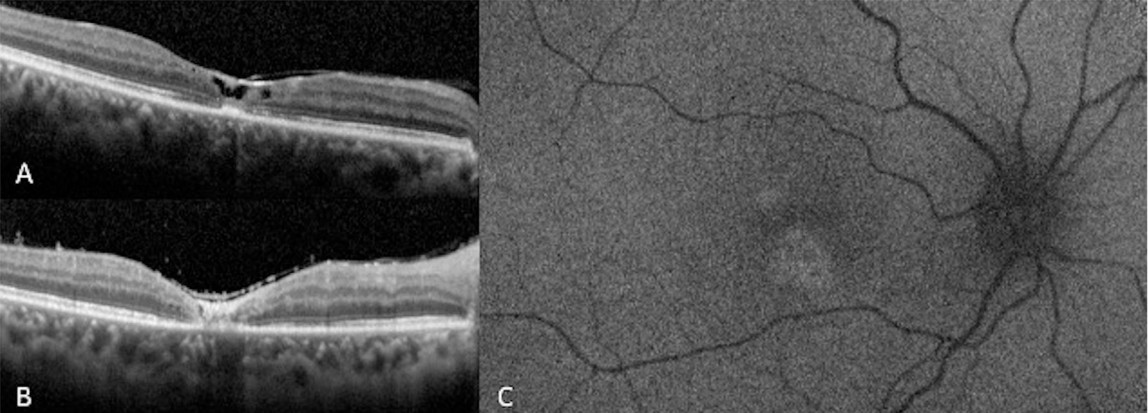

**Fig 3. Patient 3 image follow-up. A.** SD-OCT shows a PORT lesion with epiretinal membrane, full-thickness involvement, and central ellipsoid disruption. **B.** SD-OCT shows a PORT lesion after three months of treatment, retinochoroidal atrophy with backscattering corresponds to an earlier retinochoroiditis lesion. **C.** Fundus autofluorescence demonstrates mottled hypoautofluorescence of PORT lesion in perifoveal area.

## Case 6

Female in her third decade of life presented to our hospital. She referred to having blurred vision in her OD for 10 days. BCVA was 20/40 on her OD and 20/20 on her OS. IOP was 17 mmHg in OU. Clinical examination on OS showed no abnormalities. OD presented 1+cells in the anterior chamber, vitreous condensations among the temporal arcades, macular edema, and a yellowish-white punctuate perifoveal lesion with poorly defined edges. No signs of retinochoroidal scarring were seen. Toxoplasma IgG was positive (3609 IU/ml, reference value <1 IU/ml), and IgM was negative. Systemic treatment with SMZ/TMP, clindamycin, and tapering oral prednisone was initiated. Topical prednisolone was also initiated in the OD to treat anterior chamber inflammation. A month later, after completing the treatment, the patient returned with BCVA 20/25 in OU. IOP in this visit was 28 mmHg in the OD. Dorzolamide/timolol was prescribed to treat ocular hypertension. Two months later, in OU, BCVA was 20/20, showed no signs of disease activity and IOP was 13 mmHg, therefore, dorzolamide/timolol was suspended.

## Case 7

Female in her second decade of life attended for consultation. She referred progressive vision diminution in her OD for one month. BCVA is 20/200 in her OD and 20/40 in her OS. IOP were 13 and 14 mmHg in her OD and OS respectively. Clinical examination of OS showed no abnormalities. OD presented afferent pupillary defect, 2+of vitreous cells. Fundus exploration showed multiple hyperpigmented yellowish-white lesions at the macular area. SD-OCT revealed hyperreflective dots in the vitreoretinal interface, corresponding to vitritis, a subfoveal cyst, and a foveal lesion involving the outer retinal layers with underlying disruption of the ellipsoid and IZ. (**Fig 6**) Late arteriovenous phase angiography demonstrated perifoveal staining, and hyperfluorescent window defects corresponding to the chorioretinal scars. Toxoplasma IgG was elevated (80.2 IU/mL, reference value >8.8 IU/ml), and IgM was negative. SMZ/TMP, clindamycin, and tapering oral prednisone were indicated. Further follow-up was lost.

## Case 8

Female in her third decade of life presented with blurred vision in her OS for one month. BCVA is 20/20 in her OD, and 20/100 in her OS. IOP was 10 mmHg in her OD and 14 mmHg in her OS. Clinical examination of the OD showed an old toxoplasmosis scar in the inferior temporal arcade. OS had 3+cells in the anterior chamber, 2+vitreous cells,

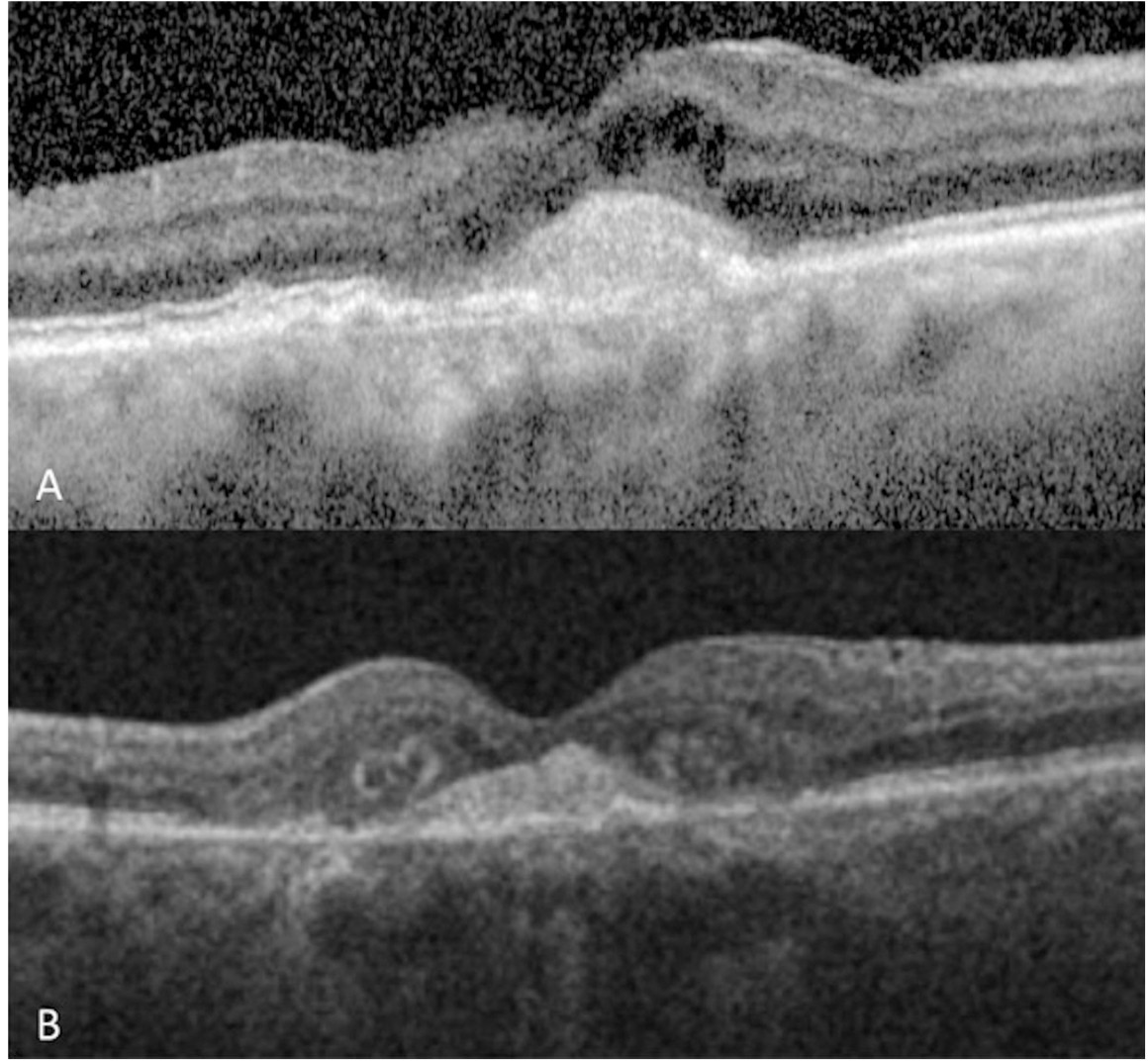

**Fig 4. SD-OCT follow-up of patient 4. A.** SD-OCT shows loss of architectural foveal with small mound of deep hyper-reflective material within zone RPE atrophy and photoreceptor disruption. **B.** SD-OCT demonstrates foveal architecture irregularity because of the presence of a smaller amount of hyper-reflective material in the fovea with RPE atrophy and photoreceptor disruption.

two chorioretinal scars, one along the superior nasal arcade, and the other one along the inferior nasal arcade, and a yellowish-white inferior temporal perifoveal lesion. Toxoplasma IgG was significantly elevated (1571 IU/mL, reference value >3 IU/ml), and IgM was negative (0.30 IU/mL, reference value >1 IU/ml). SD-OCT of this patient can be observed in **Fig 7**. Systemic treatment with SMZ/TMP, clindamycin, and tapering oral prednisone was established. OS anterior chamber inflammation was treated with topical prednisolone. Four months later the patient returned after taking the treatment for 3 months. She had a BCVA of 20/20 in her OD, and 20/30 in her OS. IOP was 19 mmHg in her OD and 18 mmHg in her OS. OD clinical examination showed no changes compared to the initial examination. OS presented no anterior chamber nor vitreous cells; the previously described chorioretinal lesions were now appreciated as scars.

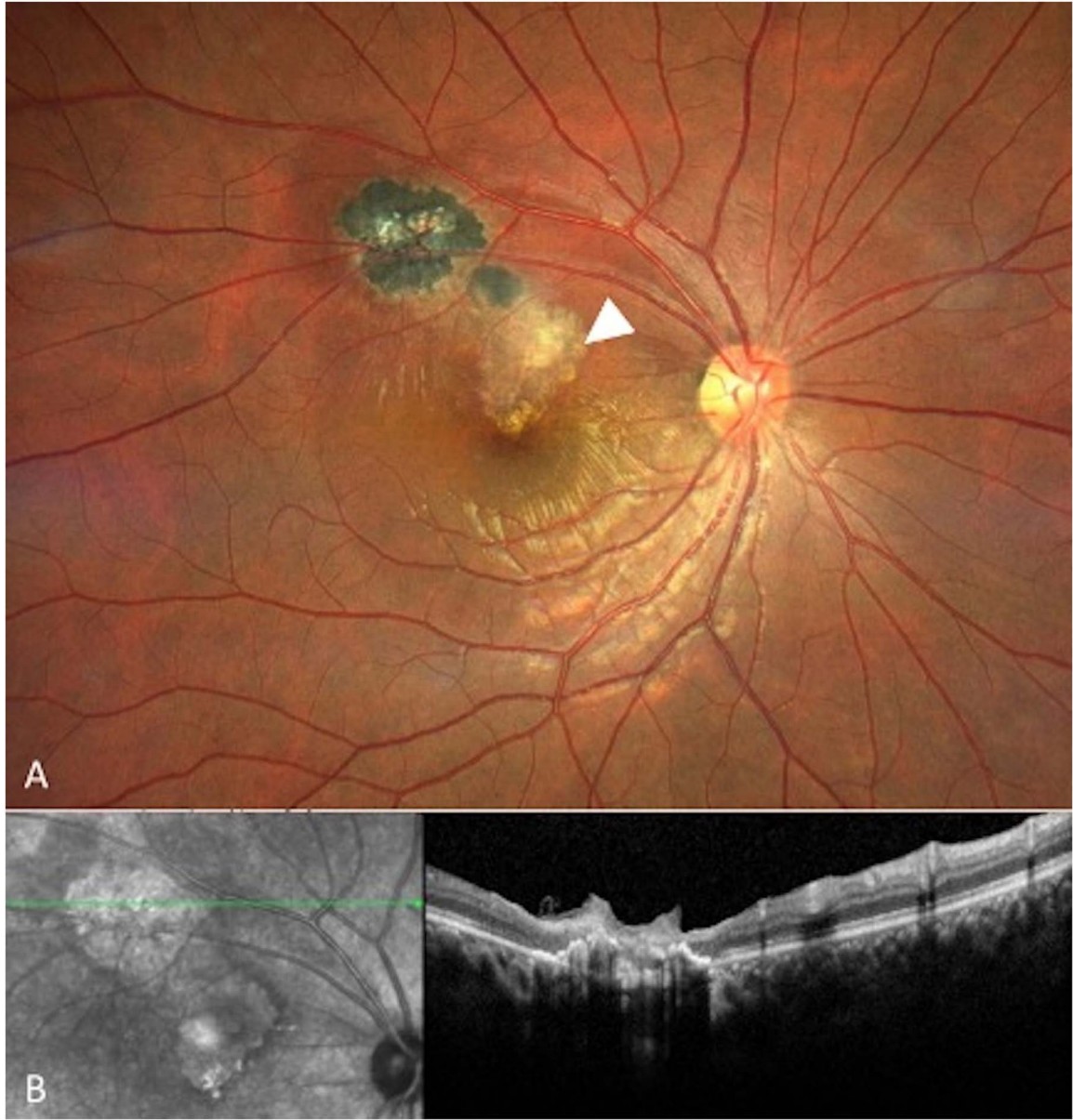

**Fig 5. A. Fundus photography of RE showing a superotemporal retinochoroidal scar and superior perifoveal white-yellowish PORT lesion (arrow). B. Green line demonstrates the location of SD-OCT image through an old chorioretinal scar with areas of irregular RPE and atrophy with visualization of Bruch's membrane.**

## Discussion

PORT is one of the atypical presentations in OT. The importance of our report relies on being the largest and most recent case series of PORT, including eight Latin-American patients with multimodal imaging follow-up and laboratory serology.

PORT has an incidence of 1.6% among OT in our hospital. An accurate global prevalence of PORT has not yet been established due to the scarcity diagnosis; however, a report made by London et al. exhibited outer deep lesions in 5% of

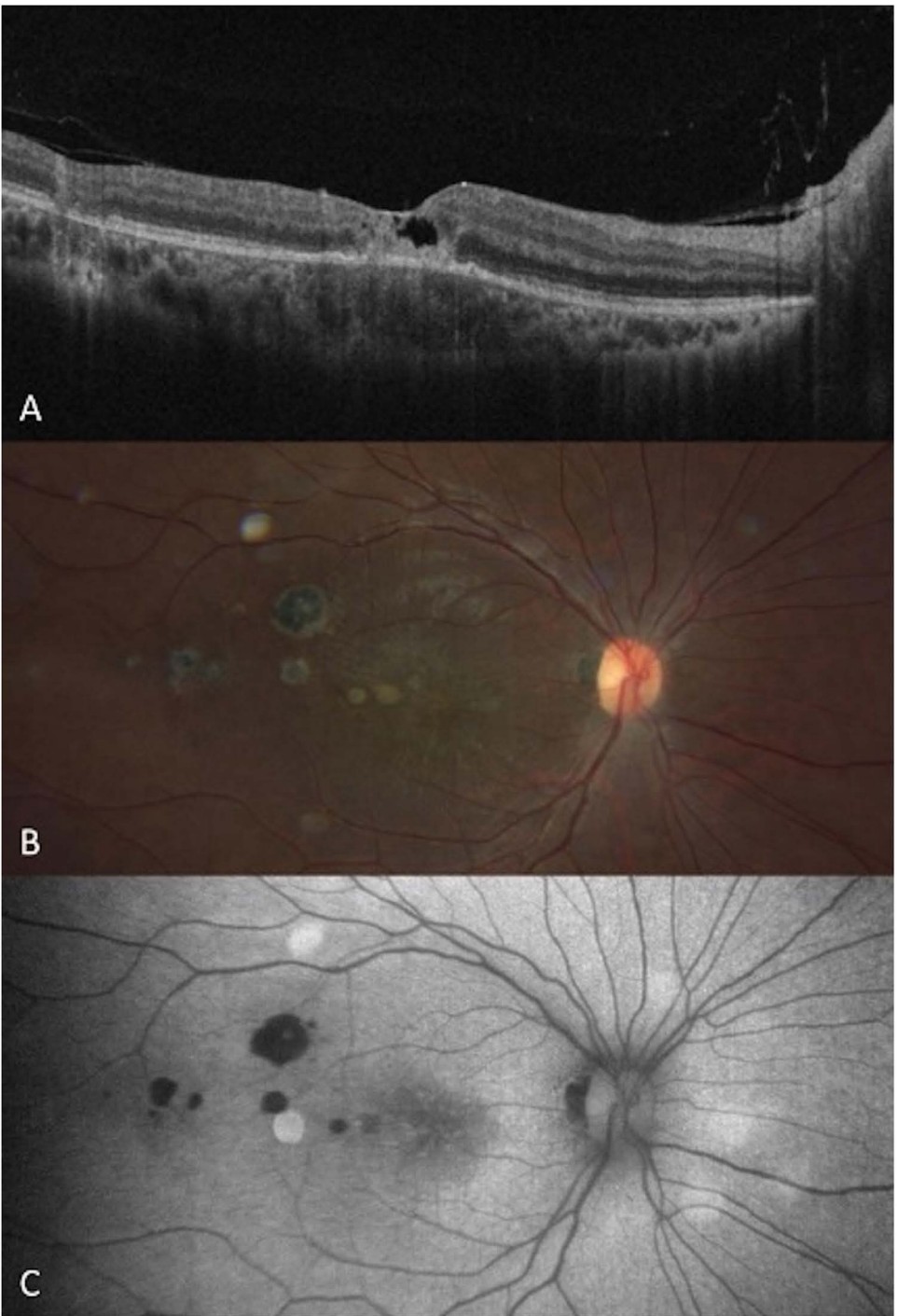

**Fig 6. Multimodal assessment of patient 7. A.** SD-OCT demonstrates a lesion with full-thickness involvement and hyper-reflective material at the outer retinal layers. **B.** Fundus photograph shows healed retinochoroiditis with pigmentation temporal to the fovea. **C.** Autofluorescence shows hypo-autofluorescence of PORT lesions.

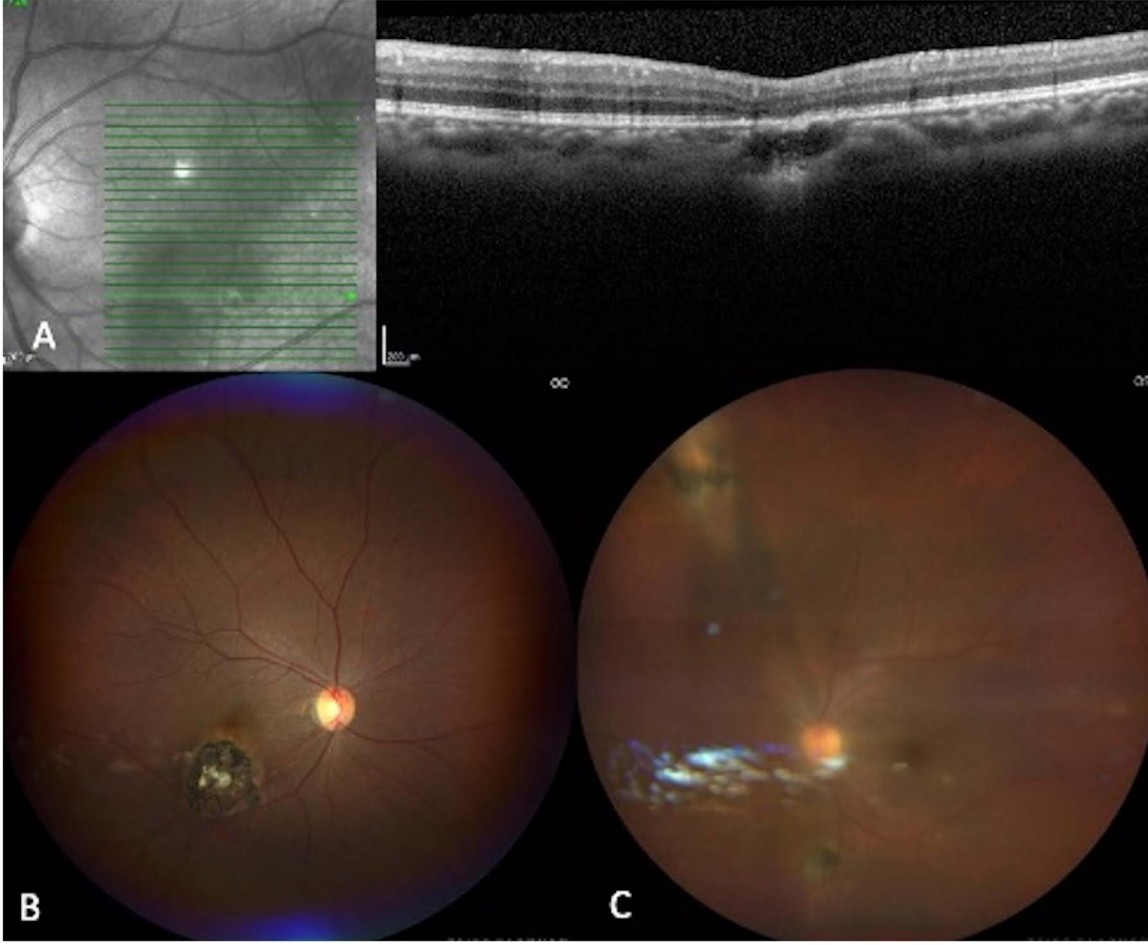

**Fig 7. Multimodal imaging of patient 8.** **A.** Green line demonstrates location of SD-OCT image: irregular thinning of the ellipsoid zone, RPE atrophy and photoreceptor disruption. **B.** Right eye widefield fundus photography shows a retinochoroidal scar between the fovea and inferotemporal arcade. **C.** Left eye widefield fundus photography shows mild vitritis, disc pallor, a retinochoroidal scar at the inferonasal arcade, and a focal yellowish lesion at the edge of a healing retinochoroiditis lesion in the superonasal area.

their cohort of OT patients. [13] Up to this date, the two largest case series are from De Souza & Casella (2009) [7], who reported five cases, and Matthews & Weiter (1988), [1] who also reported five cases.

In this study, four (50%) were de novo cases, three (37.5%) were OT reactivations, and one (12.5%) was inactive by the time the patient presented to our hospital. This highlights the possibility of PORT presenting regardless of the time-frame of toxoplasmosis infection. When assessing patients with posterior uveitis, OT always needs to be ruled out through clinical, laboratory, and imaging testing. It is also important to consider PORT as a possible first manifestation of an infection by *Toxoplasma gondii*, since small lesions can be difficult to diagnose and failure to detect them could lead to poor prognosis; we encourage to always keep PORT in mind. The diagnosis of OT is mainly presumptive based on the clinical appearance of lesions that develop at or close to the edge of an old chorioretinal scar, affecting the inner retina and causing a vitreous cellular reaction, plus serological evidence of exposure to the parasite. However, unlike classic OT, PORT is an inflammatory macular disease, that in the acute stages is characterized by multifocal grey-white lesions that appear at the level of the deep retina and retinal pigment epithelium with minimal evidence of vitritis. [11] The presence of a classical

retinochoroidal lesion does not exclude the possibility of having PORT in the same and/or the contralateral eye, as seen in patients 5 and 8. Also, even though none of our patients presented PIRT lesions during follow-up, it indeed can be found alongside PORT, therefore the presence of one should not exclude the other; careful assessment of these patients must be done do correctly diagnose atypical OT presentations.

Regarding PORT pathophysiology, several authors [4,7,11] have discussed if PORT is either a granulomatous immune response towards the toxoplasma infection or a direct cytotoxic damage made by the parasite, yet no consensus has been achieved.

To date, there is no consensus regarding non-pathological and pathological IgG ranges. [14] All patients presented increased levels of Toxoplasma Ig G with a median of 274.1 IU/dL (IQR 75.25 – 1293). Seven out of eight patients (87.5%) had negative IgM but significantly elevated IgG, hence supporting OT diagnosis. Avidity test in used to differentiate recently contracted infections from those that manifested in the past. It detects rising T. gondii specific IgG with low avidity, whereas IgM, IgA and IgE levels are usually increased. [14] The detection of high avidity antibodies is a consistent marker of chronic infection. Still, avidity test has some limitations, such as defining the level in which avidity is considered high, low or ambiguous. Additionally, avidity antibodies can remain for up to a year or more and should not be assessed as a definitive marker of a recent infection. [15]

Multimodal imaging, especially, SD-OCT, helps localize and diagnose active lesions that cause inflammation in the eye. This imaging technique shows hyperreflective inflammatory deposits at the vitreoretinal interface that extend to the inner retina and were associated with outer retinal changes including disruption of the ellipsoid zone (EZ) and interdigitation zone (IZ) and focal splitting of the retina pigment epithelium (RPE/Bruch's membrane complex). The presence of superficial retinal involvement is a distinguishing feature of this condition and aids ophthalmologists, in ruling out other diseases such as diffuse unilateral subacute neuroretinitis (DUSN), multifocal choroiditis, and punctate inner choroiditis.

OCT alterations seen in PORT could resemble some macular diseases, such as RP1L1-pathogenic-variant-related occult macular dystrophy (OMD) [16]. The most common findings in SD-OCT include disruptions or absence of the ellipsoid zone (EZ) and the interdigitation zone (IZ) [17], along with attenuation of RPE in the macular area [18]. However, OMD is characterized by a normal fundus examination [17], hence its name "occult", and it typically presents as a bilateral alteration [18,19,20]. The differential diagnosis of these conditions relies on these characteristics, as in PORT 1) the classical yellowish-white lesion is visible in fundus examination, 2) this lesion can be found outside the macular area, 3) bilateral involvement is less common. Additionally, OMD tends to be a progressive disease, whereas in PORT, visual acuity decreases more abruptly due to acute inflammation, and 4) changes in multifocal electroretinogram in OMD are always confined to the macular area [21], while in PORT, changes vary depending on where the lesion is located.

Enhanced depth imaging OCT (EDI-OCT) usually shows a mildly thickened choroid. Optical Coherence Tomography Angiography (OCT-A) may show a disrupted and enlarged foveal capillary ring. AF demonstrates hypoautofluorescence of the retinochoroidal scar and mottled perifoveal hypoautofluorescence. [3,4,12] The role of both full field electroretinogram (ffERG) and multifocal electroretinogram (mfERG) have been also studied while assessing patients with OT. Depending on the site of the lesion, mfERG could demonstrate a wide variety of alterations, this could go from local changes corresponding to the patient's scotoma in visual field examination;[22] to selective loss of responses within the central 10 degrees. [23] Norose et al. investigated the use of FERG in mice infected with OT, they concluded that ff ERG can detect site of infection, predict virulence of the organism and therefore predict the patient's prognosis. [24] Up to this date, there is no study that investigates the role of retinal electrophysiology in PORT. We believe ffERG and mfERG could be useful in patients with suspected OT that present severe vitritis when diagnosis is not clear. However, PORT is characterized by having no or mild vitritis, therefore we don't generally perform these tests in our hospital in these circumstances.

PORT is treated as the classic OT and usually responds well to therapy. [4] Given that PORT can result in poor visual recovery in some cases, we encourage the usage of oral steroids to manage retinochoroidal inflammation, thereby diminishing the chances of important retinal scar formation. Systemic steroids should be indicated at least 72 hours after

antibiotic therapy with SMZ/TMP and clindamycin, to battle both symptomatic and etiologic aspects of the disease without affecting the patient's immune response towards *Toxoplasma gondii*. [5] It is important to clarify to the patients that even though both antibiotic and anti-inflammatory treatment are being given, retinal scarring is highly probable, and final BCVA may not optimal, especially in eyes with foveal involvement. [4]

Severe inflammation in classical OT derives in ocular complications and poor visual outcomes, including raised intra-ocular pressure, cataract, retinal tear, rhegmatogenous and/or tractional retinal detachment, vascular occlusion, epireti-nal membrane, macular edema, choroidal neovascularization, vitreous haemorrhage, optic atrophy. Additionally, 18% of patients required at least one intraocular surgery. [13,14,25] PORT lesions may be small, yet require optimal management because of their macular location[4] and their significant impact on visual prognosis and patient quality of life. [3,4,11,14] Eight out of eight patients presented PORT in macular region. No PORT lesions were discovered in peripheral retina. Four patients presented PORT in the fovea (patients 1, 2, 3 and 4), two patients had perifoveal PORT lesions (patients 6 and 8) and two of them presented PORT outside foveal area but within the macula (patients 5 and 7).

In this study, we identified cases with increased intraocular pressure, choroidal neovascularization, and poor visual outcome. Four (50%) of the patients presented legal blindness (BCVA ≤ 20/200) before treatment. After treatment, three (42.8%) of the patients remained categorized as legally blind. One of the patients presenting legal blindness at the initial visit did not return, thus this data was not considered. These findings correlate with findings in a study by Bosch-Driessen et al., which reported legal blindness in at least one eye in nearly one-quarter of their subjects due to the macular location of the retinal lesions and retinal detachment. [22]

Patients need long-term follow-up since recurrences appear as PORT or as classical OT. Twenty five percent of our patients were reactivations which is less than reports of classical OT seen in up to 79% of the patients, predominantly in eyes with old scars. [22]

Eight out of eight patients were seen in the pandemic era, and we could only retrieve COVID-related information from four of them. Among these four, two had COVID-19 infections after the toxoplasmosis infection and four got a COVID vaccine after the OT diagnosis. The relationship between the prevalence of infection or reactivation of toxoplasmosis and the impact of the COVID-19 pandemic and/or COVID-19 vaccines involves many confounding factors. There may be a negative association between the prevalence of toxoplasmosis and the impact of the COVID-19 pandemic in different countries. Countries with a higher prevalence of toxoplasmosis might experience lower COVID-19 morbidity and mortal-ity, considering the following factors: 1) The prevalence of toxoplasmosis and hygiene, as it reflects the level of hygiene in a society, with lower hygiene standards and higher exposure to contaminated food or water sources. 2) Toxoplasmo-sis might have direct and immune-mediated antiviral effects, potentially protecting against viral infections. 3) A negative covariation between toxoplasmosis and COVID-19. 4) Population size and the timing of the COVID-19 epidemic might influence the relationship. 5) The wealth of nations in which these variables have been measured could bias the results. [26–30]

Our study has many strengths. We describe a case series of PORT patients, a rare disease, with adequate follow-up, imaging, and laboratory results. Additionally, it illustrates that PORT, though an infrequent presentation of OT, it can be the first episode, a recurrence, and can develop different complications. The main limitation of our study is its retrospective design; however, the unknown duration between toxoplasmosis infection and PORT presentation makes a prospective study less feasible. Also, we do not know the HIV status in 7/8 patients, and the COVID-19 infection and vaccine status in 4/8 patients.

## Conclusion

OT is the most common infectious cause of posterior uveitis. PORT is an atypical form of OT, characterized by the absence of significant vitreous cells and haze. Although PORT is not considered frequently, its complications can lead to legal blindness in a large percentage of patients who are not promptly assessed. PORT represents a diagnostic

challenge; therefore, clinical suspicion and multimodal imaging are crucial for recognizing PORT lesions. The most common and most feared complication is retinochoroidal scarring in the macular area, which can reduce a patient's BCVA to the point of legal blindness, as in 42.8% of our cases. Thus, timely and effective treatment is essential for improving visual prognosis and can potentially save the sight in these patients.

## Author contributions

**Conceptualization:** Rebeca Paquentín-Jiménez, Luz Elena Concha-del-Rio.

**Data curation:** Rebeca Paquentín-Jiménez, Ronald Rivera-Sempértegui, Luz Elena Concha-del-Rio.

**Formal analysis:** Ronald Rivera-Sempértegui, Luz Elena Concha-del-Rio.

**Investigation:** Rebeca Paquentín-Jiménez, Ronald Rivera-Sempértegui, Luz Elena Concha-del-Rio.

**Methodology:** Rebeca Paquentín-Jiménez, Ronald Rivera-Sempértegui, Luz Elena Concha-del-Rio.

**Project administration:** Luz Elena Concha-del-Rio.

**Supervision:** Luz Elena Concha-del-Rio.

**Validation:** Rebeca Paquentín-Jiménez, Luz Elena Concha-del-Rio.

**Visualization:** Luz Elena Concha-del-Rio.

**Writing – original draft:** Rebeca Paquentín-Jiménez, Luz Elena Concha-del-Rio.

**Writing – review & editing:** Luz Elena Concha-del-Rio.

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
