## [Decision Letter · Decision Letter 0]

5 Jan 2025

PNTD-D-24-00975

Small but threatening: Punctate Outer Retinal Toxoplasmosis (PORT), a case series report.

Dear Dr. Concha-del-Rio,

Thank you for submitting your manuscript to PLOS Neglected Tropical Diseases. After careful consideration, we feel that it has merit but does not fully meet PLOS Neglected Tropical Diseases's publication criteria as it currently stands. Therefore, we invite you to submit a revised version of the manuscript that addresses the points raised during the review process.

Please submit your revised manuscript within 60 days Mar 06 2025 11:59PM. If you will need more time than this to complete your revisions, please reply to this message or contact the journal office at plosntds@plos.org. Please include the following items when submitting your revised manuscript:

We look forward to receiving your revised manuscript.

Kind regards,

Aysegul Taylan Ozkan, M.D., Ph.D.,

Academic Editor

Hira Nakhasi

Section Editor

Shaden Kamhawi

co-Editor-in-Chief

Paul Brindley

co-Editor-in-Chief

**Journal Requirements:**

At this stage, the following Authors/Authors require contributions: Ronald Rivera-Sempértegui, and Luz-Elena Concha-del-Rio. Please ensure that the full contributions of each author are acknowledged in the "Add/Edit/Remove Authors" section of our submission form.

3) Some material included in your submission may be copyrighted. According to PLOSu2019s copyright policy, authors who use figures or other material (e.g., graphics, clipart, maps) from another author or copyright holder must demonstrate or obtain permission to publish this material under the Creative Commons Attribution 4.0 International (CC BY 4.0) License used by PLOS journals. Please closely review the details of PLOSu2019s copyright requirements here: PLOS Licenses and Copyright. If you need to request permissions from a copyright holder, you may use PLOS's Copyright Content Permission form.

Potential Copyright Issues:

i) Please confirm (a) that you are the photographer of 1-7, or (b) provide written permission from the photographer to publish the photo(s) under our CC BY 4.0 license.

4) We note that your Data Availability Statement is currently as follows: "Data availability can be send by the authors." Please confirm at this time whether or not your submission contains all raw data required to replicate the results of your study. Authors must share the “minimal data set” for their submission. PLOS defines the minimal data set to consist of the data required to replicate all study findings reported in the article, as well as related metadata and methods (https://journals.plos.org/plosone/s/data-availability#loc-minimal-data-set-definition).

**Comments to the Authors:**

**Please note that one of the reviews is uploaded as an attachment.**

**Reviewers' Comments:**

Reviewer's Responses to Questions

**Key Review Criteria Required for Acceptance?**

**Methods**

-Are the objectives of the study clearly articulated with a clear testable hypothesis stated?

-Is the study design appropriate to address the stated objectives?

-Is the population clearly described and appropriate for the hypothesis being tested?

-Is the sample size sufficient to ensure adequate power to address the hypothesis being tested?

-Were correct statistical analysis used to support conclusions?

-Are there concerns about ethical or regulatory requirements being met?

Reviewer #1: (No Response)

Reviewer #2: This is a case series reporting the clinical and imaging features of patients affected by punctate outer retina toxoplasmosis (PORT). The study design is appropriate and the clinical cases are well presented

Reviewer #3: Punctate outer retinal toxoplasmosis is considered a forme froste form of Ocular Toxoplasmosis. How many of the total eyes develop full thickness typical retinitis during the follow up and what was the interval between these?

One of the ways to check timeline of infection to formation of these lesions is Toxoplasma IgG avidity. Was it done in these cases?

**Results**

-Does the analysis presented match the analysis plan?

-Are the results clearly and completely presented?

-Are the figures (Tables, Images) of sufficient quality for clarity?

Reviewer #1: (No Response)

Reviewer #2: The clinical results are clearly presented. The images are appropriate and present well the clinical pictures

Reviewer #3: Macular punctate toxoplasmosis is the primary manifestation of ocular Toxoplasmosis - How many cases in the current series satisfy this criterion?

Did any of the cases have additional punctate inner retinal lesions as the combination of these lesions PIRT and PORT is a known manifestation.

**Conclusions**

-Are the conclusions supported by the data presented?

-Are the limitations of analysis clearly described?

-Do the authors discuss how these data can be helpful to advance our understanding of the topic under study?

-Is public health relevance addressed?

Reviewer #1: (No Response)

Reviewer #2: The conclusions are well supported. However, electrophysiological testing including multifocal electroretinography could have been very informative either for diagnosis or the follow-up after treatment.

Reviewer #3: Primary ocular Toxoplasmosis without old scar is often a risk factor for atypical presentation - this needs to be highlighted in the discussion section.

**Editorial and Data Presentation Modifications?**

Reviewer #1: (No Response)

Reviewer #2: Minor revisions include a discussion on the role of retinal electrophysiology, especially multifocal ERG, in the management of PORT.

Reviewer #3: Minor revision suggested

**Summary and General Comments**

Reviewer #1: (No Response)

Reviewer #2: The study is interesting and improve the knowledge in the field. I recommend minor changes.

Reviewer #3: None

PLOS authors have the option to publish the peer review history of their article (what does this mean? ). If published, this will include your full peer review and any attached files.

**Do you want your identity to be public for this peer review?** For information about this choice, including consent withdrawal, please see our Privacy Policy .

Reviewer #1: No

Reviewer #2: **Yes: ** Benedetto Falsini

Reviewer #3: No

**Figure resubmission:**
---

## [Decision Letter · Decision Letter 1]

2 Mar 2025

PNTD-D-24-00975R1Small but threatening: Punctate Outer Retinal Toxoplasmosis (PORT), a case series report.PLOS Neglected Tropical Diseases Dear Dr. Concha-del-Rio, Thank you for submitting your manuscript to PLOS Neglected Tropical Diseases. After careful consideration, we feel that it has merit but does not fully meet PLOS Neglected Tropical Diseases's publication criteria as it currently stands. Therefore, we invite you to submit a revised version of the manuscript that addresses the points raised during the review process. Please submit your revised manuscript within 30 days Apr 01 2025 11:59PM. If you will need more time than this to complete your revisions, please reply to this message or contact the journal office at plosntds@plos.org. Please include the following items when submitting your revised manuscript: * A rebuttal letter that responds to each point raised by the editor and reviewer(s). You should upload this letter as a separate file labeled 'Response to Reviewers '. This file does not need to include responses to any formatting updates and technical items listed in the 'Journal Requirements' section below. * A marked-up copy of your manuscript that highlights changes made to the original version. You should upload this as a separate file labeled 'Revised Manuscript with Track Changes '. * An unmarked version of your revised paper without tracked changes. You should upload this as a separate file labeled 'Manuscript '. If you would like to make changes to your financial disclosure, competing interests statement, or data availability statement, please make these updates within the submission form at the time of resubmission. Guidelines for resubmitting your figure files are available below the reviewer comments at the end of this letter. We look forward to receiving your revised manuscript. Kind regards, Aysegul Taylan Ozkan, M.D., Ph.D.,Academic EditorPLOS Neglected Tropical Diseases Hira NakhasiSection EditorPLOS Neglected Tropical Diseases

Shaden Kamhawi

co-Editor-in-Chief

Paul Brindley

co-Editor-in-Chief

**Journal Requirements:**

At this stage, the following Authors/Authors require contributions: Ronald Rivera-Sempértegui, and Luz-Elena Concha-del-Rio. Please ensure that the full contributions of each author are acknowledged in the "Add/Edit/Remove Authors" section of our submission form.

2) Please provide a complete Data Availability Statement in the online submission form.

3) Thank you for stating: "The authors declare no competing interests financial or non-financial, professional, or personal." Please include a completed 'Competing Interests' statement in the text box when submitting your production task, including any COIs declared by your co-authors, written in full sentences.

If you have no competing interests to declare, please state "The authors have declared that no competing interests exist". You may also provide an updated statement via email.".

**Reviewers' comments:** Reviewer's Responses to Questions

**Key Review Criteria Required for Acceptance?**

**Methods**

-Are the objectives of the study clearly articulated with a clear testable hypothesis stated?

-Is the study design appropriate to address the stated objectives?

-Is the population clearly described and appropriate for the hypothesis being tested?

-Is the sample size sufficient to ensure adequate power to address the hypothesis being tested?

-Were correct statistical analysis used to support conclusions?

-Are there concerns about ethical or regulatory requirements being met?

Reviewer #1: (No Response)

Reviewer #2: The study objectives are clearly reported, the study design is appropriate, the population clearly described and the sample size sufficiwent. Statistical analysis is correct. No ethical concerns.

Reviewer #3: The reviewer’s queries were answered adequately.

**Results**

-Does the analysis presented match the analysis plan?

-Are the results clearly and completely presented?

-Are the figures (Tables, Images) of sufficient quality for clarity?

Reviewer #1: (No Response)

Reviewer #2: The analysis marches the analysis plan. The results are clearly presented. Good quality of figures.

Reviewer #3: The reviewer’s queries were answered adequately.

**Conclusions**

-Are the conclusions supported by the data presented?

-Are the limitations of analysis clearly described?

-Do the authors discuss how these data can be helpful to advance our understanding of the topic under study?

-Is public health relevance addressed?

Reviewer #1: (No Response)

Reviewer #2: The conclusions are supported by the data and the limitations are reported. The study is novel and of public health relevance.

Reviewer #3: The reviewer’s queries were answered adequately.

**Editorial and Data Presentation Modifications?**

Reviewer #1: (No Response)

Reviewer #2: I suggest to add some comments to the discussion regarding the differential diagnosis with inherited macular dystrophies that can manifest with unilateral lesions i.e. RP1L1 gene-related macular dystrophies.

Reviewer #3: None

**Summary and General Comments**

Reviewer #1: (No Response)

Reviewer #2: This is an interesting case series describing the lesions of punctate outer retina toxoplasmosis. The clinical pictures presented in the study sometimes may be reminiscent of inherited retinal dystrophies that may not be necessarily bilateral and symmetric (see for example those related to mutations in the RP1L1 gene). The authors should discuss in more detail the differential diagnosis with inherited dystrophies.

Reviewer #3: The article could be accepted

PLOS authors have the option to publish the peer review history of their article (what does this mean? ). If published, this will include your full peer review and any attached files.

**Do you want your identity to be public for this peer review?** For information about this choice, including consent withdrawal, please see our Privacy Policy .

Reviewer #1: No

Reviewer #2: **Yes: ** Benedetto Falsini

Reviewer #3: No

**Figure resubmission:** While revising your submission, please upload your figure files to the Preflight Analysis and Conversion Engine (PACE) digital diagnostic tool, https://pacev2.apexcovantage.com/. PACE helps ensure that figures meet PLOS requirements. To use PACE, you must first register as a user. Registration is free. Then, login and navigate to the UPLOAD tab, where you will find detailed instructions on how to use the tool. If you encounter any issues or have any questions when using PACE, please email PLOS at figures@plos.org. Please note that Supporting Information files do not need this step. If there are other versions of figure files still present in your submission file inventory at resubmission, please replace them with the PACE-processed versions.**Reproducibility:** To enhance the reproducibility of your results, we recommend that authors of applicable studies deposit laboratory protocols in protocols.io, where a protocol can be assigned its own identifier (DOI) such that it can be cited independently in the future. Additionally, PLOS ONE offers an option to publish peer-reviewed clinical study protocols. Read more information on sharing protocols at https://plos.org/protocols?utm_medium=editorial-email&utm_source=authorletters&utm_campaign=protocols

---

## [Editor Report · Decision Letter 2]

26 Mar 2025

Dear Mrs Concha-del-Rio,

We are pleased to inform you that your manuscript 'Small but threatening: Punctate Outer Retinal Toxoplasmosis (PORT), a case series report.' has been provisionally accepted for publication in PLOS Neglected Tropical Diseases.

Best regards,

Aysegul Taylan Ozkan, M.D., Ph.D.,

Academic Editor

Hira Nakhasi

Section Editor

Shaden Kamhawi

co-Editor-in-Chief

Paul Brindley

co-Editor-in-Chief

---

## [Editor Report · Acceptance letter]

Dear Mrs Concha-del-Rio,

We are delighted to inform you that your manuscript, "Small but threatening: Punctate Outer Retinal Toxoplasmosis (PORT), a case series report.," has been formally accepted for publication in PLOS Neglected Tropical Diseases.

Best regards,

Shaden Kamhawi

co-Editor-in-Chief

Paul Brindley

co-Editor-in-Chief
